# Anakinra or tocilizumab in patients admitted to hospital with severe covid-19 at high risk of deterioration (IMMCoVA): A randomized, controlled, open-label trial

Jonas Sundén-Cullberg[1,2]*, Puran Chen[3], Henrike Häbel[4], Paul Skorup[5], Helena Janols[5], Johan Rasmuson[6], Katarina Niward[7], Åse Östholm Balkhed[7], Katerina Chatzidionysiou[8], Hilmir Asgeirsson[1,2], Ola Blennow[1,2], Åsa Parke[1,2], Anna-Karin Svensson[1,2], Jagadeeswara Rao Muvva[3], Hans-Gustav Ljunggren[3], Karolinska KI/K COVID-19 Treatment Working Group[¶], Anna-Carin Horne[9], Ulrika Ådén[10], Jan-Inge Henter[11], Anders Sönnerborg[1,12], Jan Vesterbacka[1,2], Piotr Nowak[1,2], Jon Lampa[8]

1 Department of Infectious Diseases, Karolinska University Hospital, Stockholm, Sweden, 2 Division of Infectious Diseases, Department of Medicine Huddinge, Karolinska Institute, Stockholm, Sweden, 3 Center for Infectious Medicine, Department of Medicine Huddinge, Karolinska Institutet, Stockholm, Sweden, 4 Division of Biostatistics, Institute of Environmental Medicine, Karolinska Institutet, Stockholm, Sweden, 5 Department of Medical Sciences, Section of Infectious Diseases, Uppsala University, Uppsala, Sweden, 6 Infection and Immunology, Department of Clinical Microbiology, Umeå University, Umeå, Sweden, 7 Department of Biomedical and Clinical Sciences, Linköping University, Linköping, Sweden, 8 Rheumatology Division, Department of Medicine Solna, Karolinska Institutet, Karolinska University Hospital, Stockholm, Sweden, 9 Theme of Children's and Women's Health, Karolinska University Hospital, Stockholm, Sweden, 10 Department of Women's and Children's Health, Karolinska Institutet, Stockholm, Sweden, 11 Childhood Cancer Research Unit, Department of Women's and Children's Health, Karolinska Institutet, Stockholm, Sweden, 12 Department of Microbiology, Tumor and Cell Biology, Karolinska Institutet, Stockholm, Sweden

¶ Membership of the Karolinska KI/K COVID-19 Treatment Working Group is listed in the Acknowledgments.
* jonas.sunden-cullberg@ki.se

**Data Availability Statement:** Relevant data for calculation of primary and secondary endpoints are

## Abstract

### Background

Anakinra and tocilizumab are used for severe Covid-19, but only one previous randomized controlled trial (RCT) has studied both. We performed a multi-center RCT comparing anakinra or tocilizumab versus usual care (UC) for adults at high risk of deterioration.

### Methods

The study was conducted June 2020 to March 2021. Eligibility required $\geq$ 5 liters/minute of Oxygen to maintain peripheral oxygen saturation at $\geq$ 93%, CRP > 70 mg/L, ferritin > 500 µg/L and at least two points where one point was awarded for lymphocytes < 1x $10^9$/L; D-dimer $\geq$ 0.5 mg/L and; lactate dehydrogenase $\geq$ 8 microkatal/L. Patients were randomly assigned 1:1:1 to receive either a single dose of tocilizumab (8 mg/kg) or anakinra 100 mg IV QID for seven days or UC alone. The primary outcome was time to recovery.

within the paper and its Supporting Information files. Data which would allow identification of individual patients such as comorbidities or inclusion date has been withheld for patient confidentiality.

**Funding:** This academic study was funded by a grant from the Swedish Research Council (2020-06318 JL;JSC;AS). JSC also received funding from Center for Innovative Medicine. The funder had no role in study design, data collection and analysis, decision to publish, or preparation of the manuscript.

**Competing interests:** The authors have declared that no competing interests exist.

## Results

Recruitment was ended prematurely when tocilizumab became part of usual care. Out of a planned 195 patients, 77 had been randomized, 27 to UC, 28 to anakinra and 22 to tocilizumab. Median time to recovery was 15, 15 and 11 days. Rate ratio for recovery for UC vs anakinra was 0.91, 0.47 to 1.78, 95% [CI], p = 0.8 and for UC vs tocilizumab 1.13, 0.55 to 2.30; p = 0.7. There were non-significant trends favoring tocilizumab (and to limited degree anakinra) vs UC for some secondary outcomes. Safety profiles did not differ significantly.

## Conclusion

Premature closure of trial precludes firm conclusions. Anakinra or tocilizumab did not significantly shorten time to clinical recovery compared to usual care. (IMMCoVA, NCT04412291, EudraCT: 2020–00174824).

## Background

Covid-19 was a leading direct and indirect cause of death worldwide during the first years of the pandemic [1]. Several treatments have become incorporated as usual care in hospitalized patients and include substances with antiviral, immunomodulatory and anti-inflammatory effects. In the latter group, current WHO recommendations for selected subgroups of patients hospitalized due to covid-19 include corticosteroids, IL-6 receptor blockers and Janus kinase (JAK) inhibitors [2]. Anakinra, an IL-1 receptor antagonist is another anti-inflammatory drug targeting the intense inflammatory response seen in many hospitalized covid-19 patients and has shown some efficacy for treatment of selected patients with severe covid-19 [3].

The Immunomodulation-CoV Assessment (ImmCoVA) multicenter study was initiated during the first wave of covid-19 in Sweden. At the time there were no published randomized controlled trials of any covid-19 therapies, but some non-randomized retrospective studies suggested a possible benefit using tocilizumab and anakinra [4'–7].

### Objectives

To evaluate the clinical efficacy and safety of tocilizumab and anakinra among hospitalized adults with laboratory-confirmed covid-19 at high risk of deterioration. At the suggestion of the Karolinska COVID-19 Treatment Working Group, we designed a randomized controlled trial comparing anakinra or tocilizumab versus usual care.

## Methods

### Trial design

ImmCoVA was a randomized, multicenter, open-label, controlled clinical trial. Enrollment began June 8, 2020, and the final patient was recruited March 16, 2021.

**Changes to trial design.** The protocol was amended after inclusion of the first two patients (S1 File).

**Participants.** Patients were eligible for enrollment if they fulfilled the following criteria: were aged ≥18 years; had laboratory-confirmed SARS-CoV-2 infection; had symptom duration of at least 7 days; required at least 5 liters/minute of supplemental oxygen for at least 8 hours, or shorter time if more than 10 liters/minute of Oxygen was needed to maintain SpO2

at ≥93%; had CRP > 70 mg/L and; Ferritin > 500 µg/L (measured up to 48 hours before inclusion); had no non-SARS-Cov2 infection, and; had at least two points on a scale of 0–3 where 1 point was awarded for each value of; lymphocytes < 1x 10(9)/L; D-dimer ≥ 0.5 mg/L and; lactate dehydrogenase ≥ 8 microkatal/L (threshold values did not have to be concurrently fulfilled and measurements made during the past 72 hours were accepted); were able to provide informed and signed consent; were willing and able to comply with study-related procedures/assessments.

Key exclusion criteria included: ongoing or completed mechanical ventilation; severe renal dysfunction eGFR < 30 mL/min; chronic liver disease; absolute neutrophil count (ANC) less than 2 x $10^9$/L, aspartate aminotransferase or alanine aminotransferase greater than 5 x upper limit of normal, platelets <100 x $10^9$/L; immunosuppression due either to preexisting autoimmune disease or ongoing treatment, acute systemic infection other than covid-19, TB, HIV or acute/chronic viral hepatitis; previous history of gastrointestinal ulceration or diverticulitis; or any physical examination findings and/or history of any illness that, in the opinion of the study investigator, might confound the results of the study or pose an additional risk to the patient by their participation in the study. The full list of exclusion criteria is presented in S1 Table.

**Study settings.** The study was conducted at four tertiary care university hospitals in Huddinge, Uppsala, Umeå and Linköping.

**Randomization and interventions.** Eligible patients were randomly assigned in a 1:1:1 ratio to receive either tocilizumab or anakinra plus usual care or usual care alone using a web-based system (www.randomize.net). Patients were block randomized with stratification by study site, sex and age. Tocilizumab was given as a single intravenous (IV) infusion of 8 mg/kg up to a maximum 800 mg. Anakinra was administered during seven days at a dose of 400 mg per day divided in doses of 100 mg IV every 6 hours. Dose was reduced in patients with renal dysfunction; in those with a creatinine clearance of 30 to 59 ml/min; the dose was 200 mg IV per day (divided in 2 doses). In patients receiving either tocilizumab or anakinra, antibiotic prophylaxis was given for seven days, either ceftriaxone 2 g once daily IV or cefotaxime 1 g x 3, alternatively, in PC allergic patients, clindamycin 600 mg x 3 IV and ciprofloxacin 500 mg x 2 orally. Other antibiotics or antibiotic combinations were also accepted, if clinically indicated. The original trial protocol prescribed antibiotic treatment in the UC arm as well, but after the first two patients had been randomized, the protocol was amended, so that patients randomized to usual care were not given antibiotic prophylaxis. One UC patient had been given AB prophylaxis per protocol before that.

All patients received usual care which was given according to national guidelines [8] and local routines.

**Procedures.** Included patients were assessed daily during hospitalization, from day 1 (day of inclusion) through day 29 and, after discharge on follow-up visits days 10, 15 and 29. Clinical status was assessed daily using an eight-category ordinal scale and the National Early Warning Score, and, at specified time points, using SOFA-score. Serious adverse events were recorded until a final follow up at 60 days.

**Outcomes.** To facilitate comparisons with other studies, we used the same primary endpoint and ordinal scale as in the pivotal remdesivir trial (ACCT-1) [9] which was a slightly modified version of that proposed by the WHO. The primary outcome was the time to recovery, defined as the first day, during day 1–29 after enrollment, on which a patient met the criteria for category 1, 2, or 3 on the eight-category ordinal scale. The categories were as follows:

1, not hospitalized; 2, not hospitalized, with limitation of activities and/or requiring home oxygen; 3, hospitalized, not requiring supplemental oxygen and no longer requiring ongoing medical care; 4, hospitalized, not requiring supplemental oxygen but requiring ongoing

medical care (related to covid-19 or to other medical conditions); 5, hospitalized, requiring supplemental oxygen; 6, hospitalized, requiring noninvasive ventilation or use of high-flow nasal cannula; 7, hospitalized, on invasive mechanical ventilation or ECMO; and 8, death.

Time to recovery was calculated as days elapsed from randomization until day of recovery.

Secondary outcomes included mortality by day 29, length of hospital stay, elevated level of care (ICU or high dependency unit), days on supplemental oxygen and utilization of mechanical, non-invasive ventilation or high frequency nasal cannula at any time-point and mean number of ventilator-free days alive. Degree of improvement was evaluated on three assessment scales: mean improvement in the 8-point ordinal scale from baseline and nadir value days 1–7 until days 8 and 15, and mean improvement in SOFA and NEWS2 scores from baseline until days 5, 10 and 15. We compared fractions of patients who had: achieved ordinal scale 1–4 by day 15 and who were discharged to institution other than normal domicile. The safety population was defined according to initiation of either study drug or usual care only. Safety outcomes included incidence of: serious adverse events; severe or life-threatening infections; infusion reactions.

The cumulative dose of steroids was calculated for each patient. Dosages of methylprednisolone, hydrocortisone, prednisone and dexamethasone were calculated separately and converted to betamethasone equivalents. Use of remdesivir was registered.

**Statistical analysis.** The statistical analysis plan was finalized prior to recruitment of the first patient. Sample size was calculated based on two approaches: the classical conventional approach and the adaptive design approach.

Related to clinical experience and pre-trial data the usual care group was expected to reach the primary outcome after a mean of 15 +/- 6 days. Based on published literature and off-label experience of tocilizumab in Karolinska University Hospital Huddinge, we expected the time for reaching the primary outcome to decrease to 11 days in the anakinra and tocilizumab treatment arms.

Taking the significance level at 5% and Type II error as 20% and an expected mean difference of 4 days, assuming a standard deviation of 6, 36 subjects were required per group to obtain 80% power. Given an anticipated dropout rate of 10%, a total of 40 subjects per group allocated 1:1:1 to UC, anakinra and tocilizumab (120 patients in all) was needed to obtain 80% power.

Since the available data on covid-19 and potential effects of interventions to date was sparse, we expected that small variations in the magnitude of difference between intervention groups and control group in reaching the primary outcome could result in decreased power of the analysis. Therefore, an adaptive design was employed, allowing re-estimation of sample size at an interim point, when about 50% of the patients had been included. The interim analysis was performed by a statistician blinded to treatment assignment after recruitment of 74 patients. It found larger variance in the primary outcome than anticipated, 60 and 63 in the study arms with the largest between-group difference of mean days to recovery. This indicated that at least 61 patients had to be included per arm to maintain power. Given possible dropouts, decision was taken to increase sample size to 195 (65 patients per arm).

The data were summarized as means (standard deviations (SD)) for symmetrically distributed numerical variables and medians (interquartile ranges (IQR)) for numerical variables with skewed distributions or categorical ordinal variables. Positive counts and percentages were presented for binary variables. Missing data were handled as missing at random.

Analyses were performed by intention-to-treat, i e patients were analyzed as randomized. For the primary analysis of time to recovery, rate ratios stratified by time were calculated using the Mantel-Cox method. A corresponding log-rank test was conducted comparing anakinra or tocilizumab with usual care. Kaplan-Meier estimates of cumulative recoveries were

estimated. Time was censored for patients who died at their day of death. For patients who had not recovered by then, time was censored at day 29. In a sensitivity analysis, hazard ratios were estimated using Cox proportional hazard regression models either crude, adjusted for age and sex or adjusted for age, sex and BMI.

For the secondary analysis, differences in mortality were assessed using the Chi-squared test. Median time-to-event and confidence intervals were estimated. Differences in median time-to-event were calculated and confidence intervals estimated using bootstrap. Rate ratios were estimated and compared as in the primary analysis. Confidence intervals for proportions or pairwise differences of proportions were obtained and percentage point differences evaluated under a binomial distribution assumption. Confidence intervals for number of days or pairwise difference in mean number of days were estimated and differences evaluated according to the assumptions of a t-test. In sensitivity analyses of the main outcome, crude age and sex adjusted regression analyses were conducted using a Cox proportional hazard model. The purpose of the sensitivity analyses was to explore potential effects of age and sex on hazard ratios, between treatment groups due to the discontinuation of the recruitment.

A significance level of 0.05 was chosen and two-sided p-values were reported. The robust sandwich estimator was used to estimate standard errors for all regression models. All analyses were conducted in STATA version 16.1 (StataCorp, College Station, TX).

**Data and safety monitoring board.** An independent data and safety monitoring board reviewed results after 56 patients had been recruited and approved continued inclusion.

*Ethical section*. The trial protocol was approved by the Swedish Medical Products Agency, and the ethical review board in Stockholm (no 2020–01973, 2020–02530, 2020–0482, 2020–05709), and was overseen by an independent data and safety monitoring board. All patients received oral and written information and signed consent forms. Details concerning trial design, oversight and analyses and protocol amendments are available in S1 and S4 Files.

## Results

During the study, tocilizumab was adopted as part of usual care at which point recruitment was discontinued. By that time, 77 out of an intended 195 patients had been randomized, 27 to usual care and 28 and 22 to anakinra and tocilizumab, respectively (Fig 1). All patients received assigned therapy, except two allotted to anakinra who both received study drug for four instead of seven days, for details refer to S2 File. Both were included in the anakinra group in analyses. All patients were followed up for 60 days or until death and all were included in intention-to-treat analyses.

Mean age of study participants was 61 and 78% were male (Table 1). Recruitment started later at three of four planned sites. One hospital (Karolinska) included 74 patients, three were included from the other sites (two from Uppsala, one from Umeå, none from Linköping). Of all patients, 29% had no significant medical history, 36% had hypertension and 21% had diabetes, 46% were past or current smokers, median BMI was 28. Median SOFA score was 4 and median NEWS2 was 6. Median number of days between onset of symptoms and randomization was 9 (IQR 8–11).

### Severity of disease

As specified by inclusion criteria, all patients fulfilled the WHO definition of severe disease, *i.e.* required oxygen; 30 (39%) fulfilled category five criteria on the ordinal scale, 46 (59%) category six and one patient (1%) category 7. In the UC cohort, 71% of patients belonged to the more severe categories 6 or 7 at baseline vs 50% in the anakinra and 64% in the tocilizumab group (p = 0.1 for UC vs anakinra, p = 0.6 for UC vs tocilizumab). Baseline NEWS2 was higher in the

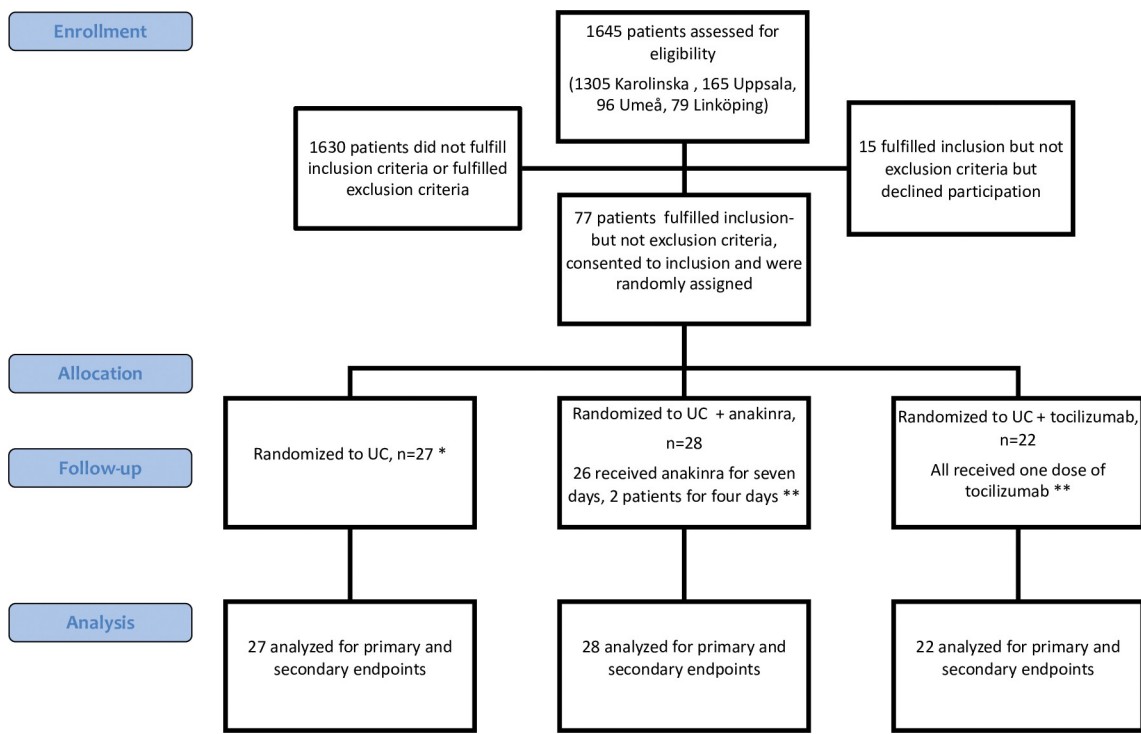

**Fig 1. CONSORT flow diagram.** UC = usual care. * One patient received prophylactic antibiotics before protocol amendment. ** All patients received prophylactic antibiotics for seven days.

control group than in either treatment group (p = 0.01 for UC vs anakinra, p = 0.03 for UC vs tocilizumab).

## Primary outcome

Time to recovery was a median 15 (10–19) 95% confidence interval [CI], 15 (9–24) and 11 (10–19) days in the UC, anakinra and tocilizumab groups, respectively. The differences between UC vs anakinra and UC vs tocilizumab were not statistically significant. Rate ratio for recovery for UC vs anakinra was 0.91 95% confidence interval [CI], 0.47 to 1.78, p = 0.8 and for UC vs tocilizumab 1.13; 95% [CI], 0.55 to 2.30; p = 0.7, Table 2 and Fig 2. In a sensitivity analysis we adjusted for age and sex. *Unadjusted* hazard ratios were 0.94 95% [CI], (0.54 to 1.67) for UC vs anakinra and 1.08 95% [CI], (0.60 to 1.97) for UC vs tozilizumab. Corresponding figures in an analysis adjusted for age and sex were 1.08 95% [CI], (0.60 to 1.95) for UC vs anakinra and 1.24 95% [CI], (0.68 to 2.26) for UC vs tozilizumab. Ordinal score changes over time are illustrated in Fig 3.

## Secondary outcomes

Secondary endpoints included mortality up to day 29, length of hospital stay, level of care, changes in oxygen requirement/delivery and changes in severity measured by ordinal scale, NEWS2 and SOFA scores. In summary, there were no significant differences between UC and anakinra or between UC and tocilizumab for any measure (Table 2). It should be noted, however, that for some important outcomes there were trends signaling better effect in the treatment groups, mainly the tocilizumab group. These were: time to hospital discharge among survivors (16, 13 and 11 days in the UC, anakinra and tocilizumab groups respectively); days

**Table 1. Patient characteristics at baseline by randomised groups.**

| | Usual care | Anakinra | Tocilizumab | All patients |
|---|---|---|---|---|
| **No of patients** | 27 | 28 | 22 | 77 |
| **Age (mean, SD)** | 61 ± 12 | 61 ± 13 | 63 ± 10 | 61 ± 12 |
| **Sex male n (%)** | 19/27 (70%) | 22/28 (79%) | 19/22 (86%) | 60/77 (78%) |
| **BMI (median, IQR)** | 28 (26, 31) | 28 (24, 34) | 28 (28, 34) | 28 (26, 32) |
| **PaO2/FiO2 ratio kPa (median, IQR)** | 15 (13, 23) * | 16 (11, 23) * | 16 (12–20) ** | 15 (12, 21) **** |
| **CRP mg/L (median, IQR)** | 134 (81, 191) | 137 (106, 185) | 152 (110, 180) * | 139 (101, 190) * |
| **PCT µg/L (median, IQR)** | 0.26 (0.14, 0.48) | 0.23 (0.19, 0.54) * | 0.36 (0.16, 0.54) * | 0.30 (0.16, 0.53), ** |
| **Ferritin µg/L (median, IQR)** | 1687 (958, 2568) | 1400 (1003, 2353) | 1398 (903, 2967) | 1429 (968, 2430) |
| **IL-6 ng/L (median, IQR)** | 26 (10, 53) * | 30 (15, 44) | 24 (12–92) ** | 26 (11, 56) *** |
| **D-Dimer mg/L FEU (median, IQR)** | 1.04 (0.6, 1.6) * | 0.88 (0.54, 1.25) | 1.20 (0.94–1.42) * | 1.03 (0.59, 1.47) ** |
| **LD µkatal/L (median, IQR)** | 7.9 (6.6, 10.3) *** | 7.9 (6.1, 9.3) | 9.5 (8.0, 10.2) * | 8.1 (6.6, 9.8) **** |
| **Lymphocytes x 10⁹/L (median, IQR)** | 0.8 (0.5, 0.8) | 0.7 (0.5, 0.8) | 0.7 (0.6–0.9) * | 0.7 (0.5–0.9) * |
| **Bilirubin µmol/l. (median, IQR)** | 8 (6, 11) | 8 (6, 11) | 10 (8, 12) * | 8 (6, 12) * |
| **History of AMI, n (%)** | 1 (4%) | 2 (7%) | 2 (9%) | 5 (6%) |
| **CHF, n (%)** | 0 | 0 | 1 (5%) | 1 (1%) |
| **Peripheral arterial disease, n (%)** | 0 | 0 | 2 (9%) | 2 (3%) |
| **Previous CVL or TIA, n (%)** | 3 (11%) | 2 (7%) | 0 | 5 (6%) |
| **Hypertension, n (%)** | 10 (37%) | 11 (39%) | 7 (32%) | 28 (36%) |
| **Diabetes, n (%)** | 6 (22%) | 5 (18%) | 5 (23%) | 16 (21%) |
| **COPD, n (%)** | 1 (4%) | 0 | 1 (5%) | 2 (3%) |
| **Asthma, n (%)** | 2 (7%) | 3 (11%) | 2 (9%) | 7 (9%) |
| **Previous/current smoker, n (%)?** | 11 (41%) | 19 (36%) | 10 (45%) * | 35 (46%) * |
| **SOFA (median, IQR)** | 4 (3, 5) | 3.5 (3, 4) | 4 (3, 4) | 4 (3, 4) |
| **SOFA mean (SD)** | 3.9 ± 1.4 | 3.6 ± 1.0 | 3.8 ± 1.1 | 3.8 ± 1.1 |
| **NEWS2 (median, IQR)** | 7 (6,9) | 5.5 (3.5,7) | 6 (4,7) | 6 (5,8) |
| **NEWS2 mean, (SD)** | 7.0 ± 2.0 | 5.5 ± 2.1 | 5.7 ± 2.0 | 6.1 ± 2.1 |
| **Ordinal scale, median (IQR)** | 6 (5,6) | 5.5 (5,6) | 6 (5,6) | 6 (5,6) |
| **Ordinal scale mean, (SD)** | 5.7 ± 0.5 | 5.5 ± 0.5 | 5.6 ± 0.5 | 5.6 ± 0.5 |
| **Ordinal scale category 5, requiring supplemental oxygen, n (%)** | 8 (30%) | 14 (50%) | 8 (36%) | 30 (39%) |
| **Ordinal scale category 6, NIV/HFNC, n (%)** | 18 (67%) | 14 (50%) | 14 (64%) | 46 (59%) |
| **Ordinal scale category 7, Mechanical Ventilation, n (%)** | 1 (4%) | 0 | 0 | 1 (1%) |
| **Median days elapsed from symptom onset to randomization** | 10 (10, 12) | 9 (8, 11) | 9 (8, 11) | 9 (8, 11) |
| **Highest level of care on day of randomization** | | | | |
| **ED/Ward** | 15 (55%) | 16 (57%) | 13 (59%) | |
| **HDU** | 2 (7%) | 5 (18%) | 3 (14%) | |
| **ICU** | 10 (37%) | 7 (25%) | 6 (27%) | |

* one missing.

** two missing

*** three missing

**** four missing. AMI = Acute myocardial infarction, BMI = Body mass index, CHF = Congestive heart failure, COPD = Chronic obstructive pulmonary disease, CVL = Cerebrovascular lesion, CRP = C–reactive protein, ED = Emergency department, FEU = Fibrinogen equivalent units, HDU = High dependency unit, HFNC = High flow nasal cannula, ICU = Intensive care unit, IL–6 = Interleukin 6, IQR = Interquartile range, LD = Lactate dehydrogenase, NEWS2 = National Early Warning Score 2, NIV = Non–invasive ventilation, PaO2/FiO2 = Partial pressure of oxygen/fraction of inspired oxygen, PCT = Procalcitonin, SD = Standard deviation, SOFA = Sequential Organ Failure Assessment, TIA = Transitory ischemic attack

Table 2. Primary and secondary outcomes.

| Primary outcome | Usual care, n = 27 | Anakinra, n = 28 | Tocilizumab, n = 22 | UC vs anakinra | UC vs Tocilizumab |
|---|---|---|---|---|---|
| Median days until recovery, (95% confidence interval) | 15 (10 to 19) | 15 (9 to 24) | 11 (10 to 19) | Rate ratio 0.91 (0.47 to 1.78), p = 0.8 | 1.13 (0.55 to 2.32), p = 0.7 |
| **Secondary outcomes** | | | | | |
| **Mortality** | | | | | |
| By day 29 n (%) | 2 (7%) | 2 (7%) | 2 (9%) | p = 1.0 | p = 0.9 |
| By day 60 | 3 (11%) | 3 (11%) | 2 (9%) | p = 1.0 | p = 0.8 |
| **Hospitalisation** | | | | | |
| Median days to initial hospital discharge (category 7, 8 on the ordinal scale) 95% [CI] | 16 (11 to 24) | 16 (8 to.) | 12 (10 to 28) | 0 (-9 to 11) Days difference, 95% [CI] | -4 (-8 to 11) Days difference, 95% [CI], |
| Median days to initial hospital discharge (category 7, 8 on the ordinal scale) in patients *who survived for* 28 days. 95% [CI] | 16 (11 to 20) | 13 (8 to 26) | 11 (10 to 14) | -3 (-9 to 10) Days difference, 95% [CI] | -5 (-10 to 0) Days difference, 95% [CI] |
| **Oxygen** | | | | | |
| Days with supplemental oxygen, 95% [CI] | 14 (10 to 25) | 11 (7 to 28) | 10 (9 to 18) | -3 (-9 to 7) Days difference, 95% [CI] | -4 (-8 to 3) Days difference, 95% [CI] |
| Mechanical ventilation at any point during the 28 first days, %, 95% [CI] | 22 (9 to 42) | 11(2 to 28) | 14 (3 to 35) | -11 (-31 to 8) Percentage points difference, 95% [CI] | -8 (-30 to 13) Percentage points difference, 95% [CI] |
| Patients who used NIV at any point during the 28 first days, % 95% [CI] | 37% (19 to 58) | 46% (28 to 66) | 27% (11 to 50) | 9% (-17 to 35) Percentage points difference, 95% [CI] | - 10% (-36 to 16) Percentage points difference, 95% [CI] |
| Patients who used HFNC at any point during the 28 first days, % 95% [CI] | 81% (62 to 94) | 82 (46%) (63 to 94 | 73 (27%) (50 to 89) | 1% (-20 to 21) Percentage points difference, 95% [CI] | - 9% (-33 to 14) Percentage points diffence, 95% [CI] |
| Mean number of ventilator-free days alive, days, 95% [CI] | 25.2 (22.5 to 27.8), | 26.3 (24.5 to 28.2), | 26.1 (23.3 to 29.0), | 1.1 (-2.0 to 4.3), Days difference, 95% [CI] | 0.9 (-2.9 to 4.8) Days difference, 95% [CI] |
| **Improvement** | | | | | |
| Median days until improvement of one category on ordinal scale, 95% [CI] | 8 (6 to 11) 95% [CI] | 8 (7 to 10) | 9 (6 to 11) | 0.9 (0.5 to 1.8) #Rate Ratio 95% [CI] | 0.9 (0.4 to 1.7) #Rate Ratio 95% [CI] |
| **Mean improvement in the 8-point ordinal scale from baseline** | | | | | |
| until day 8, days, 95% [CI] | -0.9 (-1.6 to -0.1) | -0.7 (-1.2 to -0.2) | -0.6 (-1 to -0.1) | 0.2 (-0.4 to 0.8), Points difference, 95% [CI] | 0.3 (-0.4 to 0.9), Points difference, 95% [CI] |
| until day 15, days, 95% [CI] | -2.3 (-3.1 to -1.4) | -2.3 (-3.0 to -1.5) | -3.1 (-4.1 to -2.2) | 0.0 (-0.8 to 0.8), Points difference, 95% [CI] | -0.9 (-2.1 to 0.3), Points difference, 95% [CI] |
| **Mean improvement between nadir ordinal score day 1–7** | | | | | |
| until day 8, days, 95% [CI] | -1.2 (-1.8 to -0.6) | -1.0 (-1.6 to -0.5) | -0.8 (-1.2 to -0.3) | 0.1 (-0.4 to 0.7), Points difference, 95% [CI] | 0.4 (-0.2 to 1.1), Points difference, 95% [CI] |
| until day 15, days, 95% [CI] | -2.6 (-3.4 to -1.8) | -2.6 (-3.4 to -1.8) | -3.3 (-4.2 to -2.4) | 0.0 (-0.8 to. 0.7), Points difference, 95% [CI] | -0.7 (-1.8 to 0.3), Points difference, 95% [CI] |
| **Achieved ordinal scale 1–4** | | | | | |
| by day 15, (%), 95% [CI] | 52% (32 to 71) | 54% (34 to 72) | 73% (50 to 89) | 2 (-25 to 28) Points difference (95% CI) | 21 (-6 to 47) Points difference (95% CI) |
| **Mean improvement in SOFA score from baseline until days 5, 10 and 15:** | | | | | |
| Day 5 95% [CI] | 0 (-0.8 to 0.8) | -0.4 (-0.9 to 0.2) | -0.5 (-1.2 to 0.1) | -0.4 (-1.1 to 0.4), Points difference, 95% [CI] | -0.5 (-1.4 to 0.3), Points difference, 95% [CI] |
| Day 10 95% [CI] | -1.6 (-2.4 to -0.8) | -1.4 (-2.1 to -0.7) | -1.7 (-2.4 to -1.0) | 0.2 (0.5 to 1.0), Points difference, 95% [CI] | -0.1 (-0.8 to 0.7), Points difference, 95% [CI] |
| Day 15 95% [CI] | -1.9 (-2.9 to -1.0) | -1.5 (-2.5 to -0.5) | -2.1 (-2.9 to -1.3) | 0.4 (-0.6 to 1.4), Points difference, 95% [CI] | -0.2 (-1.0 to 0.7), Points difference, 95% [CI] |
| **Mean improvement in NEWS2 score from baseline until days 5, 10 and 15** | | | | | |
| Day 5 95% [CI] | -1.8 (-3.1 to -0.5) | -0.9 (-2.1 to 0.3) | -0.3 (-1.5 to 0.9) | 0.9 (-0.5 to 2.4), Points difference, 95% [CI] | 1.6 (-0.4 to 3.5), Points difference, 95% [CI] |

*(Continued)*

**Table 2.** (Continued)

| Primary outcome | Usual care, n = 27 | Anakinra, n = 28 | Tocilizumab, n = 22 | UC vs anakinra | UC vs Tocilizumab |
|---|---|---|---|---|---|
| Day 10 95% [CI] | -3.0 (-4.1 to -1.9) | -2.5 (-3.7 to -1.2) | -1.9 (-(-3.0 to -0.8) | 0.5 (-0.7 to 1.8), Points difference, 95% [CI] | 1.1 (-0.4 to 2.6), Points difference, 95% [CI] |
| Day 15 95% [CI] | -3.3 (-4.7 to -2.0) | -2.4 (-3.8 to -1.1) | -2.6 (-4.1 to -1.1) | 0.9 (-0.6 to 2.5), Points difference, 95% [CI] | 0.7 (-0.8 to 2.1), Points difference, 95% [CI] |
| **Admitted at any time point during hospital stay to the:** | | | | | |
| ICU Percent of patients, 95% [CI] | 48 (29 to 68) | 50 (31 to 69) | 36 (17 to 59) | 2 (-25 to 28), Percentage points difference, 95% [CI] | - 12 (-39 to 16) Percentage points difference, 95% [CI] |
| HDU and/or ICU Percent of patients 95% [CI] | 56 (35 to 75) | 64 (44 to 81) | 50 (28 to 72) | 9 (-17 to 35) Percentage points difference, 95% [CI] | - 6 (-34 to23) Percentage points difference, 95% [CI] |
| Patients released to rehabilitation by day 29, percent of patients, 95% [CI] | 11 (2 to 29) | 21 (8 to 41) | 9 (1 to 29) | 10 (-9 to 30) Percentage points difference, 95% [CI] | - 2 (-19 to 15) Percentage points difference, 95% [CI] |

CI = Confidence interval, HDU = High dependency unit, HFNC = High flow nasal cannula, ICU = Intensive care unit, NEWS2 = National Early Warning Score 2, NIV = Non–invasive ventilation, SD = Standard deviation, SOFA = Sequential Organ Failure Assessment, UC = Usual care

on supplemental oxygen (14, 11 and 10); fraction of patients on mechanical ventilation (22, 11 and 14%) and admitted to ICU (48, 50 and 36%) and; degree of improvement by day 15 on the ordinal scale (-2.3, -2.3 and -3.1). Improvement on the NEWS2 scale by day 15, on the other hand, was most pronounced in the UC group (-3.3, -2.4, and -2.6).

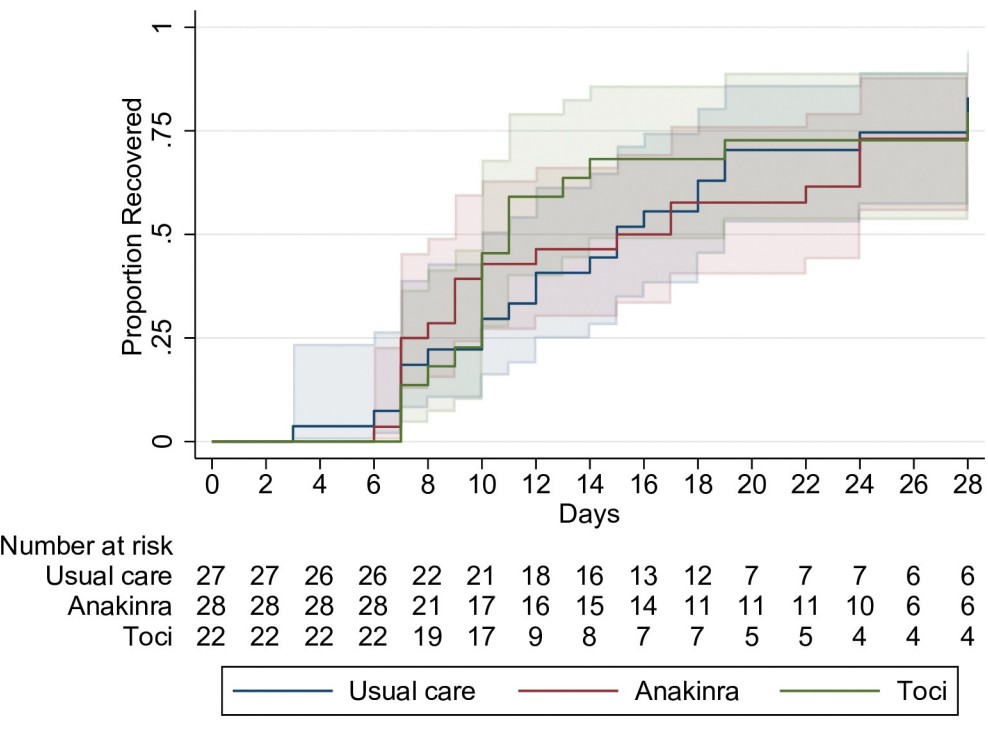

**Fig 2. Proportion of patients recovered per treatment allocation day 1–29, Kaplan Meier.**

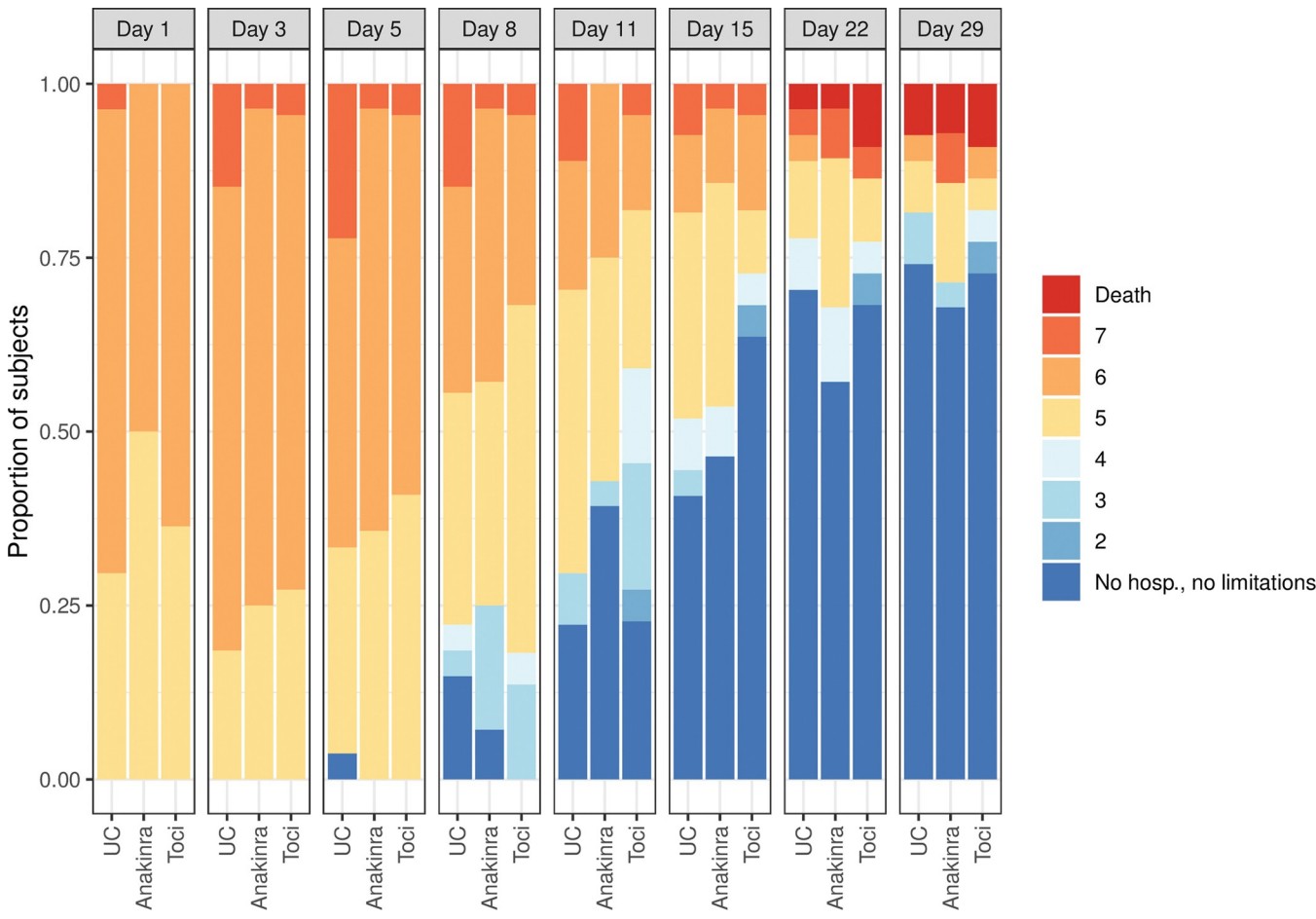

**Fig 3. Ordinal score changes per treatment allocation.**

### Steroid and remdesivir use

Two patients (one in the UC arm and one in the anakinra arm) were recruited before cortico-steroids became part of usual care, the rest received steroids. The cumulative dose of steroids during the study (no patient received steroids after day 29) was a total betamethasone equivalents of 1929 mg in the UC arm, 2412 in the anakinra arm and 1742 in the tocilizumab arm. Median totals of betamethasone equivalents for patients who received steroids were 71 (48–88), 65 (46–80) and 74 (52–102) mg respectively. Remdesivir was administered to 4 (15%), 3 (11%) and 5 (23%) patients in the UC, anakinra and tocilizumab arms.

### Adverse events

Serious adverse events occurred on or before day 29 in 13 patients (48%) in the UC, 14 (50%) in the anakinra and in 6 (27%) in the tocilizumab arm, Table 3 and S2 Table, non-significant differences. There were 6 deaths up to day 29, two in each arm. Two additional patients died on or after d29, please see S3 File for details. Infections occurred in 3, 2 and 1 patient(s) in the UC, anakinra and tocilizumab arms. For further data on adverse events, please refer to S3 Table.

**Table 3. Serious adverse events by system organ class and treatment arm by study day 29.**

|  | Usual Care n = 27 | Anakinra n = 28 | Tocilizumab n = 22 |
|---|---|---|---|
| **Any system** | 34 | 26 | 17 |
| **Death** | 2 | 2 | 2 |
| **Gastrointestinal disorders** | 2 | 1 | 0 |
| **Infections and Infestations^** | 4 | 3 | 1 |
| **Cardiac disorders** | 1 | 0 | 0 |
| **Surgical and medical procedures** | 1 | 0 | 0 |
| **Laboratory abnormalities** | 0 | 0 | 2 |
| **Respiratory, thoracic and mediastinal disorders\*** | 16 | 15 | 10 |
| **Renal and urinary disorders** | 1 | 1 | 0 |
| **Nervous system disorders** | 2 | 0 | 0 |
| **Psychiatric disorders** | 1 | 0 | 0 |
| **Vascular disorders** | 1 | 1 | 0 |
| **General disorders and administration site conditions** | 3 | 3 | 2 |

Number of patients with any SAE: 13 in UC group, 14 in Anakinra group and 6 in Tocilizumab group. Some patients had more than one SAE.

^Infections: UC group: 1 Staphylococcus aureus septicaemia and 1 ventilator–associated pneumonia (same patient), 1 bacterial pneumonia, 1 post intubation tracheitis. In the anakinra group: 1 Streptococcus mitis bacteremia and 1 E Coli bacteremia (same patient), 1 Post–covid deterioration. In the tocilizumab group: 1 nosocomial infection of unknown origin.

\*10 in the UC group had respiratory failure/progress of respiratory failure. 3 pulmonary emboli, 1 pneumothorax, 1 aspiration pneumonia. 1 pulmonal hypertension. In the anakinra group: 13 had respiratory failure/progress of respiratory failure, 1 pulmonary embolus, 1 pneumothorax. In the tocilizumab group: 7 had respiratory failure/progress of respiratory failure, 1 pneumomediastinum, 1 lung fibrosis, 1 pulmonal hypertension

## Discussion

This was an open-label, randomized, controlled, multi-center trial comparing anakinra or tocilizumab versus usual care in patients with severe covid-19 at high risk of deterioration. Recruitment of patients was discontinued prematurely when decision, due to emerging data from the REMAP-CAP and RECOVERY trials [10, 11], was taken to include tocilizumab as part of usual care. The primary and secondary efficacy analyses in our study did not show any significant differences related to treatment allocation, nor any difference in occurrence of adverse events. However, since only 77 out of a planned 195 patients had been recruited by study closure, statistical power was insufficient to preclude either positive or negative effects from the study drugs.

Although differences between usual care and the two treatments were not statistically significant for any measure, time to recovery, the primary endpoint, was numerically shorter in the tocilizumab group compared to the usual care arm. Trends favoring tocilizumab were also seen with fewer severe adverse advents and for several secondary endpoints, including fewer days with supplemental oxygen, shorter time to hospital discharge, lower proportion of patients admitted to ICU and receiving mechanical ventilation, and a higher proportion achieving ordinal scale 1–4 by day 15. Improvements in ordinal score also showed tendencies toward better numbers for tocilizumab at day 15, but not at earlier time points. There were limited corresponding positive trends for anakinra.

Mortality by 29 and 60 days was similar and unexpectedly low in all three treatment arms, despite inclusion criteria designed to select for higher severity, and despite half of patients being admitted at some point to either ICU or high dependency unit. Baseline severity, as measured by NEWS2, was higher in the usual care group (but only marginally higher as assessed by SOFA and ordinal scores) which complicates assessment. Consequently, improvement in

NEWS2 by days 5, 10 and 15 was numerically (but non-significantly) higher in the usual care group.

Although some early RCTs using IL-6 blockade failed to show clinical effect [12, 13], the REMAP-CAP and RECOVERY platform trials [10, 11] demonstrated improved survival which was also confirmed in a subsequent metanalysis of 27 RCTs [14] (including this study). Tocilizumab has been incorporated as a therapeutic option in usual care alongside the JAK inhibitor baricitinib–another immune-modifying drug which has been shown to reduce covid-19 mortality [15, 16] and which is recommended by WHO for use in severe covid-19 either alone, or in combination with tocilizumab [2]. Few peer-reviewed RCTs have been published on anakinra. One study found no effect [17] and another was terminated early because of safety concerns [18]. Only one RCT has demonstrated improved outcome. It used levels of soluble urokinase plasminogen activator receptor–suPAR to risk-stratify patients–only those with elevated concentrations were included. Patients who were randomized to receive anakinra had lower odds of progression of COVID-19 on an 11-point WHO Clinical Progression Scale (median values differed one point) [3]. Anakinra has been approved for use by the European Medicines Agency (EMA), but uptake will likely be limited since use is conditioned on elevated plasma levels of suPAR, an analysis which is not widely available, and which introduces a diagnostic delay. Also, the question remains whether anakinra has any benefit over tocilizumab or baricitinib. In the present study, although underpowered, there were no convincing tendencies to suggest an effect of anakinra superior to usual care. To our knowledge, only one other prospective study–COV-AID–has published data on concurrent IL-1 and IL-6 blockade [19]. It used a factorial design in which the same patients were randomized twice, first 1:2 to anakinra: no anakinra, then 1:1:1 to siltuximab (another IL-6 blocking agent): tocilizumab: no IL-6 blockade. Outcomes were compared between treatment and control within each randomization group and no differences were found in the primary outcome of time to clinical improvement. Possible explanations for lack of effect were discussed–COV-AID used standard dose of anakinra *i.*e. 100 mg subcutaneously once daily, and the authors speculated that this may have been insufficient. In our study, standard dosing was 100 mg IV four times daily with weak to no signs of effect, suggesting that either even higher doses might be required or that anakinra makes small or no difference. The COV-AID authors also hypothesized that a lack of demonstrable effect compared with previous studies could be due to differences in severity of disease in the study populations. Mortality in the usual care group was 10%, on par with 7% in the present study, and much lower than 33 and 35% in the REMAP-CAP and RECOVERY trials. Other studies have shown that the most pronounced effects of interleukin blockage are found in patients with high oxygen requirement and risk of death [2]. Demonstrating modest benefits for patients with lower illness severity and low event rates requires large studies. Major challenges for future trials of drugs targeting inflammation in covid-19 will thus be to identify patients that are sufficiently ill so that potential drug benefits will be large enough to be demonstrable in study populations of manageable size. The fraction of severely ill patients among those hospitalized is currently shrinking fast. Disease outcome is improving for many reasons–rising immunity after vaccinations and previous infection(s), viral evolution benefiting strains with lower virulence, better covid-19 drugs and generally improved management thanks to increased familiarity with the disease.

## Limitations

The relatively few patients included precludes firm conclusions from this study. Other limitations include the open-label design. This was chosen in part because tocilizumab use is hard to mask given its normalizing effects on fever and CRP, in part because placebo procedures

would have taken too long to set in motion. All patients were recruited at tertiary university hospitals, the absolute majority from Karolinska University hospital in south Stockholm, which limits external validity. The proportion of patients that received remdesivir as part of usual care was highest in the tocilizumab group and lowest in the anakinra group, but since duration of illness per inclusion criteria was at least seven days, and since remdesivir likely has minimal effect when administered late in immunocompetent patients with covid-19, this is unlikely to have affected outcome. Patients were continuously assessed for eligibility, but due to high workload, reasons for non-eligibility were not registered.

## Conclusions

In this study in patients with covid-19 and respiratory failure, anakinra or tocilizumab did not significantly shorten time to clinical improvement compared to usual care. Safety profiles did not differ. However, premature closure of trial and fewer included patients than planned precludes firm conclusions.

## Supporting information

**S1 Table. Inclusion and exclusion criteria.**
(DOCX)

**S2 Table. Severe adverse events by day 60.**
(DOCX)

**S3 Table. Adverse events by day 60.**
(DOCX)

**S4 Table. Consort checklist.**
(PDF)

**S1 File. Study protocol ImmCoVA v.4.0.**
(PDF)

**S2 File. Protocol amendments during study.**
(DOCX)

**S3 File. Recruitment interruption and adherence to protocol.**
(DOCX)

**S4 File. Causes of death.**
(DOCX)

**S5 File. Study data.**
(XLSX)

## Acknowledgments

We are grateful to the Karolinska KI/K COVID-19 Treatment Working Group for providing the impetus for this study. The working group medical specialties and members were as follow: in alphabetic orders: Hematology: Martin Jädersten; Infectious diseases: Andreas Berge, Piotr Nowak, Anders Sönnerborg, Jan Vesterbacka; Intensive care: Lars I Eriksson, Anders Oldner; Pediatrics: Tatiana von Bahr Greenwood, Jan-Inge Henter (coordinator), AnnaCarin Horne, Kim Ramme; Rheumatology: Francesca Faustini, Lars Klareskog, Jon Lampa.

We are also very grateful to research nurses Linn Wursé and Ann-Marie Strömberg who collected samples, followed up patients and recorded data in the clinical research form after inclusion and to Cecilia Lång och Johanna Löfberg who recorded data. The greatest thanks of all to research nurse Anna Hollander who provided expertise during study design and tirelessly worked as main organizer of the research nurses during the study as well as performing large parts of follow-up procedures herself.

## Author Contributions

**Conceptualization:** Jonas Sundén-Cullberg, Helena Janols, Anna-Carin Horne, Ulrika Ådén, Jan-Inge Henter, Anders Sönnerborg, Jan Vesterbacka, Piotr Nowak, Jon Lampa.

**Data curation:** Jonas Sundén-Cullberg, Puran Chen.

**Formal analysis:** Jonas Sundén-Cullberg, Henrike Häbel, Jon Lampa.

**Funding acquisition:** Jon Lampa.

**Investigation:** Jonas Sundén-Cullberg, Paul Skorup, Johan Rasmuson, Katarina Niward, Åse Östholm Balkhed, Katerina Chatzidionysiou, Hilmir Asgeirsson, Ola Blennow, Åsa Parke, Anna-Karin Svensson, Jan Vesterbacka, Piotr Nowak.

**Project administration:** Jonas Sundén-Cullberg, Jagadeeswara Rao Muvva, Hans-Gustav Ljunggren, Jon Lampa.

**Resources:** Hans-Gustav Ljunggren.

**Writing – original draft:** Jonas Sundén-Cullberg.

**Writing – review & editing:** Henrike Häbel, Anna-Carin Horne, Jon Lampa.

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
