## [Decision Letter · Decision Letter 0]

19 Jul 2023

PONE-D-23-07456Anakinra or tocilizumab in patients admitted to hospital with severe covid-19 at high risk of deterioration (IMMCoVA): a randomized, controlled, open-label trialPLOS ONE

Dear Dr. Sunden-Cullberg,

Thank you for submitting your manuscript to PLOS ONE. After careful consideration, we feel that it has merit but does not fully meet PLOS ONE’s publication criteria as it currently stands. Therefore, we invite you to submit a revised version of the manuscript that addresses the points raised during the review process. Please submit your revised manuscript by Sep 01 2023 11:59PM. If you will need more time than this to complete your revisions, please reply to this message or contact the journal office at plosone@plos.org. Please include the following items when submitting your revised manuscript:A rebuttal letter that responds to each point raised by the academic editor and reviewer(s). You should upload this letter as a separate file labeled 'Response to Reviewers'.A marked-up copy of your manuscript that highlights changes made to the original version. You should upload this as a separate file labeled 'Revised Manuscript with Track Changes'.An unmarked version of your revised paper without tracked changes. You should upload this as a separate file labeled 'Manuscript'.If applicable, we recommend that you deposit your laboratory protocols in protocols.io to enhance the reproducibility of your results. Protocols.io assigns your protocol its own identifier (DOI) so that it can be cited independently in the future. For instructions see: https://journals.plos.org/plosone/s/submission-guidelines#loc-laboratory-protocols. Additionally, PLOS ONE offers an option for publishing peer-reviewed Lab Protocol articles, which describe protocols hosted on protocols.io. Read more information on sharing protocols at https://plos.org/protocols?utm_medium=editorial-email&utm_source=authorletters&utm_campaign=protocols.

We look forward to receiving your revised manuscript.

Kind regards,

Avik Ray

Academic Editor

PLOS ONE

Journal Requirements:

“This academic study was funded by a grant from the Swedish Research Council (https://www.vr.se/english.html)(JL;JSC;AS). The council had no part in data collection, analyses or interpretation, nor in writing of the manuscript, or decision to submit.”

Reviewers' comments:

Reviewer's Responses to Questions

**Comments to the Author**

1. Is the manuscript technically sound, and do the data support the conclusions?

Reviewer #1: Yes

2. Has the statistical analysis been performed appropriately and rigorously? 

Reviewer #1: Yes

3. Have the authors made all data underlying the findings in their manuscript fully available?

Reviewer #1: No

4. Is the manuscript presented in an intelligible fashion and written in standard English?

Reviewer #1: Yes

5. Review Comments to the Author

Reviewer #1: This is an interesting RCT assessing the clinical efficacy and safety of tocilizumab and anakinra among hospitalized adults with laboratory-confirmed covid-19 at high risk of deterioration.

Some main points to consider:

Randomisation was stratified by study site, sex and age. Understandably the recruitment was discontinued leading to an impact in imbalance, however they seems to be imbalance between groups for sex. Was there adjustment done for this in the analysis?

The study shows that in addition an adaptive design approach was implemented for the re-adjustment of sample size at 50% of subjects recruited and an interim analysis performed. Suggest the authors also include the pre-specified alpha spent at this look at the data. This should be mentioned in the methods.

Other minor point to consider for clarification of the manuscript.

1. In the sample size calculation statement (line 199) include the SD (assumed to be 6 when recalculating the sample size)

2. In the statistical analysis section - define your population of analysis i.e ITT since this is stated in the results section. i.e analysed as they were randomised etc..

3. Define your safety population - e.g those who took at least one dose of treatment or started treatment etc..

4. Suggest to rename table 1 to "Table 1 Patient characteristics at baseline by randomised groups"

5. This is an open label study, mention finalization of the statistical analysis plan i.e prior to first patient recruitment was this done?

6. Mention handling of missing data in the stats section

---

## [Author Response · Author response to Decision Letter 0]

30 Aug 2023

Please find our point-by-point responses to requests made by PLOS ONE and to questions and suggestions by the reviewer. The information below is the same as in the document resposes to reviewers. Our responses below are in CAPITAL LETTERS. 

a) Please clarify the sources of funding (financial or material support) for your study. List the grants or organizations that supported your study, including funding received from your institution. THE FUNDER (SWEDISH RESEARCH COUNCIL) HAS BEEN LISTED.

b) State what role the funders took in the study. If the funders had no role in your study, please state: “The funders had no role in study design, data collection and analysis, decision to publish, or preparation of the manuscript.” THE TEXT HAS BEEN CHANGED ACCORDINGLY, LINE 448-449..

c) If any authors received a salary from any of your funders, please state which authors and which funders. NO SALARY WAS PAID BY FUNDERS.

d) If you did not receive any funding for this study, please state: “The authors received no specific funding for this work.” NOT APPLICABLE

Please include your amended statements within your cover letter; we will change the online submission form on your behalf. WE HAVE SPECIFIED CHANGES IN THE COVER LETTER.

3. We note that the grant information you provided in the ‘Funding Information’ and ‘Financial Disclosure’ sections do not match. THE TEXT NOW READS: “THIS ACADEMIC STUDY WAS FUNDED BY A GRANT FROM THE SWEDISH RESEARCH COUNCIL (2020-06318 JL;JSC;AS). JSC ALSO RECEIVED FUNDING FROM CENTER FOR INNOVATIVE MEDICINE.” LINE 469-470. HOWEVER, WE HAVEN´T FIGURED OUT HOW TO CHANGE TO THE SAME TEXT IN THE FINANCIAL DISCLOSURE SECTION. 

When you resubmit, please ensure that you provide the correct grant numbers for the awards you received for your study in the ‘Funding Information’ section. THE GRANT NUMBER HAS BEEN ADDED TO THE MANUSCRIPT. LINE 447

4. In your Data Availability statement, you have not specified where the minimal data set underlying the results described in your manuscript can be found. PLOS defines a study's minimal data set as the underlying data used to reach the conclusions drawn in the manuscript and any additional data required to replicate the reported study findings in their entirety. All PLOS journals require that the minimal data set be made fully available. For more information about our data policy, please see http://journals.plos.org/plosone/s/data-availability. THE DATA SET HAS NOW BEEN ADDED AS STUDY DATA - SUPPORTING INFORMATION FILES.

5. Your ethics statement should only appear in the Methods section of your manuscript. If your ethics statement is written in any section besides the Methods, please move it to the Methods section and delete it from any other section. Please ensure that your ethics statement is included in your manuscript, as the ethics statement entered into the online submission form will not be published alongside your manuscript. THE ETHICS STATEMENT HAS BEEN MOVED TO THE METHOD SECTION.

6. Please review your reference list to ensure that it is complete and correct. If you have cited papers that have been retracted, please include the rationale for doing so in the manuscript text, or remove these references and replace them with relevant current references. Any changes to the reference list should be mentioned in the rebuttal letter that accompanies your revised manuscript. If you need to cite a retracted article, indicate the article’s retracted status in the References list and also include a citation and full reference for the retraction notice. WE HAVE REVIEWED THE REFERENCE LIST AND BELIEVE THAT IT IS COMPLETE AND CORRECT. WE FOUND NO RETRACTED PAPERS IN THE LIST.

Reviewers' comments:

Reviewer's Responses to Questions

FIRST, THANK YOU FOR A CAREFUL REVIEW AND VALUABLE COMMENTS, QUESTIONS AND SUGGESTIONS WHICH HAVE RESULTED IN CHANGES THAT WE BELIEVE HAS IMPROVED THE MANUSCRIPT! 

Comments to the Author

1. Is the manuscript technically sound, and do the data support the conclusions?

Reviewer #1: Yes

2. Has the statistical analysis been performed appropriately and rigorously? 

Reviewer #1: Yes

3. Have the authors made all data underlying the findings in their manuscript fully available?

Reviewer #1: No

4. Is the manuscript presented in an intelligible fashion and written in standard English?

Reviewer #1: Yes

5. Review Comments to the Author

Reviewer #1: This is an interesting RCT assessing the clinical efficacy and safety of tocilizumab and anakinra among hospitalized adults with laboratory-confirmed covid-19 at high risk of deterioration.

THANK YOU!

Some main points to consider:

Randomisation was stratified by study site, sex and age. Understandably the recruitment was discontinued leading to an impact in imbalance, however they seems to be imbalance between groups for sex. Was there adjustment done for this in the analysis? NO, ALL ANALYSES WERE UNADJUSTED. WE HAVE ADDED A SENSITIVITY ANALYSIS ADJUSTED FOR SEX AND AGE FOR THE PRIMARY OUTCOME, LINES 224-226 AND 312-316.

The study shows that in addition an adaptive design approach was implemented for the re-adjustment of sample size at 50% of subjects recruited and an interim analysis performed. Suggest the authors also include the pre-specified alpha spent at this look at the data. This should be mentioned in the methods. SINCE THIS WAS A BLINDED ANALYSIS FOR THE SOLE PURPOSE OF RECALCULATING SAMPLE SIZE, WE CONSIDERED IT TO HAVE LIMITED EFFECT ON THE TYPE I ERROR, AND THAT THE INTERIM ANALYSIS THUS SPENT NO ALPHA. . 

Other minor point to consider for clarification of the manuscript.

1. In the sample size calculation statement (line 199) include the SD (assumed to be 6 when recalculating the sample size). Thank you for pointing this out. THE TEXT HAS BEEN AMENDED AND NOW READS “TAKING THE SIGNIFICANCE LEVEL AT 5% AND TYPE II ERROR AS 20% AND AN EXPECTED MEAN DIFFERENCE OF 4 DAYS, WHILE ASSUMING A STANDARD DEVIATION OF 6, 36 SUBJECTS WERE REQUIRED PER GROUP TO OBTAIN 80% POWER. “ LINE 201.

2. In the statistical analysis section - define your population of analysis i.e ITT since this is stated in the results section. i.e analysed as they were randomised etc..THE TEXT HAS BEEN AMENDED AND NOW READS “ANALYSES WERE PERFORMED BY INTENTION-TO-TREAT, I E ANALYSED AS RANDOMIZED.” LINE 216

3. Define your safety population - e.g those who took at least one dose of treatment or started treatment etc. The text has been amended and now reads THE TEXT HAS BEEN AMENDED AND NOW READS “THE SAFETY POPULATION WAS DEFINED ACCORDING TO INITIATION OF EITHER STUDY DRUG OR USUAL CARE ONLY” LINE 183-184

4. Suggest to rename table 1 to "Table 1 Patient characteristics at baseline by randomised groups" THANK YOU FOR THE SUGGESTION. THE TABLE HAS BEEN RENAMED. LINE 270.

5. This is an open label study, mention finalization of the statistical analysis plan i.e prior to first patient recruitment was this done? The following has been specified: “THE STATISTICAL ANALYSIS PLAN WAS FINALIZED PRIOR TO RECRUITMENT OF THE FIRST PATIENT.” LINE 192.

6. Mention handling of missing data in the stats section. THIS INFORMATION HAS BEEN ADDED: “MISSING DATA WERE HANDLED AS MISSING AT RANDOM.” LINE 218

---

## [Decision Letter · Decision Letter 1]

1 Dec 2023

Anakinra or tocilizumab in patients admitted to hospital with severe covid-19 at high risk of deterioration (IMMCoVA): a randomized, controlled, open-label trial

PONE-D-23-07456R1

Dear Dr. Sunden-Cullberg,

We’re pleased to inform you that your manuscript has been judged scientifically suitable for publication and will be formally accepted for publication once it meets all outstanding technical requirements.

Kind regards,

Avik Ray

Academic Editor

PLOS ONE

Additional Editor Comments (optional):

Reviewers' comments:

Reviewer's Responses to Questions

**Comments to the Author**

1. If the authors have adequately addressed your comments raised in a previous round of review and you feel that this manuscript is now acceptable for publication, you may indicate that here to bypass the “Comments to the Author” section, enter your conflict of interest statement in the “Confidential to Editor” section, and submit your "Accept" recommendation.

Reviewer #1: All comments have been addressed

Reviewer #2: (No Response)

2. Is the manuscript technically sound, and do the data support the conclusions?

Reviewer #1: Yes

Reviewer #2: Yes

3. Has the statistical analysis been performed appropriately and rigorously? 

Reviewer #1: Yes

Reviewer #2: Yes

4. Have the authors made all data underlying the findings in their manuscript fully available?

Reviewer #1: No

Reviewer #2: Yes

5. Is the manuscript presented in an intelligible fashion and written in standard English?

Reviewer #1: Yes

Reviewer #2: No

6. Review Comments to the Author

Reviewer #1: (No Response)

Reviewer #2: This study is an interesting RCT that evaluates the clinical effectiveness and safety of tocilizumab and anakinra in hospitalized adults with confirmed COVID-19 who are at a high risk of deterioration.

Since the treatment protocols for COVID-19 were very dynamic, can you please define the usual care in your RCT in the methods?

You mentioned, "During the study, tocilizumab was adopted as part of usual care at which point recruitment was discontinued", Would you consider your study to be against this? Please comment on this to provide a clear implication of this RCT.

7. PLOS authors have the option to publish the peer review history of their article (what does this mean?). If published, this will include your full peer review and any attached files.

Reviewer #1: No

Reviewer #2: No

---

## [Editor Report · Acceptance letter]

18 Dec 2023

PONE-D-23-07456R1 

PLOS ONE

Dear Dr. Sunden-Cullberg, 

I'm pleased to inform you that your manuscript has been deemed suitable for publication in PLOS ONE. Congratulations! Your manuscript is now being handed over to our production team.

Kind regards, 

on behalf of

Dr. Avik Ray 

Academic Editor

PLOS ONE